# Relationship between Photosynthetic CO_2_ Assimilation and Chlorophyll Fluorescence for Winter Wheat under Water Stress

**DOI:** 10.3390/plants12193365

**Published:** 2023-09-23

**Authors:** Qianlan Jia, Zhunqiao Liu, Chenhui Guo, Yakai Wang, Jingjing Yang, Qiang Yu, Jing Wang, Fenli Zheng, Xiaoliang Lu

**Affiliations:** 1College of Natural Resources and Environment, Northwest A&F University, Xianyang 712100, China; qianlanjia@nwafu.edu.cn (Q.J.); guochenhui@nwafu.edu.cn (C.G.); wangyakai@nwafu.edu.cn (Y.W.); 2State Key Laboratory of Soil Erosion and Dryland Farming on the Loess Plateau, Northwest A&F University, Xianyang 712100, China; zliu@nwafu.edu.cn (Z.L.); yuq@nwafu.edu.cn (Q.Y.); wangjing.je@163.com (J.W.); flzh@ms.iswc.ac.cn (F.Z.); 3The Research Center of Soil and Water Conservation and Ecological Environment, Chinese Academy of Sciences and Ministry of Education, Xianyang 712100, China; yangjingjing20@mails.ucas.ac.cn; 4Institute of Soil and Water Conservation, Chinese Academy of Sciences and Ministry of Water Resources, Xianyang 712100, China; 5College of Resources and Environment, University of Chinese Academy of Sciences, Beijing 100049, China

**Keywords:** photosynthesis model, photosynthetic CO_2_ assimilation, pulse-amplitude modulation (PAM), remote sensing, solar-induced chlorophyll fluorescence (SIF), water stress

## Abstract

Solar-induced chlorophyll fluorescence (SIF) has a high correlation with Gross Primary Production (GPP). However, studies focusing on the impact of drought on the SIF-GPP relationship have had mixed results at various scales, and the mechanisms controlling the dynamics between photosynthesis and fluorescence emission under water stress are not well understood. We developed a leaf-scale measurement system to perform concurrent measurements of active and passive fluorescence, and gas-exchange rates for winter wheat experiencing a one-month progressive drought. Our results confirmed that: (1) shifts in light energy allocation towards decreasing photochemistry (the quantum yields of photochemical quenching in PSII decreased from 0.42 to 0.21 under intermediate light conditions) and increasing fluorescence emissions (the quantum yields of fluorescence increased to 0.062 from 0.024) as drought progressed enhance the degree of nonlinearity of the SIF-GPP relationship, and (2) SIF alone has a limited capacity to track changes in the photosynthetic status of plants under drought conditions. However, by incorporating the water stress factor into a SIF-based mechanistic photosynthesis model, we show that drought-induced variations in a variety of key photosynthetic parameters, including stomatal conductance and photosynthetic CO_2_ assimilation, can be accurately estimated using measurements of SIF, photosynthetically active radiation, air temperature, and soil moisture as inputs. Our findings provide the experimental and theoretical foundations necessary for employing SIF mechanistically to estimate plant photosynthetic activity during periods of drought stress.

## 1. Introduction

Light energy absorbed by plants is consumed in three competing pathways: photochemistry (photochemical quenching (PQ)), emission of chlorophyll *a* fluorescence (ChlF), and non-photochemical quenching (NPQ) [1]. ChlF emission is the radiative loss of absorbed solar energy in the spectral range from 640 nm to 850 nm, with emission peaks at 685 nm and 740 nm [2]. NPQ is the process by which plants dissipate absorbed photon energy as heat and consists of two components: basal or constitutive heat dissipation (D), and regulated heat dissipation (N). Energy partitioning between these three pathways may be highly dynamic under changing physiological and environmental conditions [3,4]. Thus, by measuring ChlF, one may obtain valuable information on the other two processes, namely, PQ and NPQ. Active ChlF measurements, mainly based on the active pulse-amplitude modulation (PAM) technique [5], have been successfully used to assess changes in photosynthetic machinery and the photosynthetic status of plants [6,7]. However, they are generally applied at the leaf scale because active manipulation of the light environment is required [3]. In recent years, substantial progress has also been made in the passive detection of solar-induced chlorophyll fluorescence (SIF) within solar or telluric absorption features (e.g., O_2_-A at 760 nm), enabling the top-of-canopy (TOC) SIF observations (SIF*_toc_*) in discrete and narrow wavelength bands to be obtained from remote sensing platforms [8,9,10]. Many studies have shown that canopy SIF has a strong capacity to predict gross primary productivity (GPP) across a variety of land cover types [4,11,12,13]. Previous studies have indicated that GPP and SIF have a strong linear relationship, and usually exhibit similar spatial and temporal patterns [9,14,15]. Marrs et al. (2020) [16] suggested that the linear relationship between SIF and GPP at large spatial and temporal scales is the result of a shared driver.

As a consequence of climate change, droughts are expected to increase in frequency, duration and severity in many parts of the world, most notably in Africa, Asia and Central and South America [17], and the drought-induced reduction in crop yields has received widespread attention [18,19,20,21]. It is noteworthy that winter wheat is the one of the most important and widely planted staple crops in the world [22], and the normal growth and development of winter wheat can be influenced by drought stress [23,24,25]. Therefore, a rigorous evaluation of the performance of SIF in detecting early signs of photosynthetic downregulation during droughts is particularly relevant to food security. However, SIF-based studies in this field have reported inconsistent results: satellite or near-ground SIF measurements show remarkable declines during droughts [26,27,28], but leaf-level SIF/ChlF measurements have a weak and delayed response to water stress [29,30]. Moreover, a key step in estimating GPP from remotely sensed SIF is to parameterize the SIF-GPP relationship with data-driven statistical approaches: a combination of flux-tower measurements and satellite SIF data are used. These approaches do not usually track photosynthesis at regional or global scales, particularly facing the complexity of naturally varying systems. All these controversies and limitations highlight the urgent need to develop a practical approach for estimating GPP. This approach should be based on a mechanistic understanding of the relationship between ChlF and photosynthetic CO_2_ assimilation under drought conditions.

To obtain a complete picture of the mechanisms regulating ChlF/SIF emission and photosynthetic CO_2_ assimilation under a water deficit, one should not only measure passive ChlF radiance for drought plants, but also, simultaneously, gas exchange and active PAM fluorescence. In this study, we developed a concurrent leaf-level measurement system consisting of a portable gas-exchange system, a PAM instrument, and four high-resolution spectrometers. Using this measurement system, we obtained a variety of key photosynthetic parameters, actively and passively induced ChlF, and the light–response curves of gas exchange for both drought-affected and control plants during a month-long progressive drought experiment. We first show the cascade of decline in these photosynthetic parameters, which include stomatal conductance, net photosynthetic carbon assimilation, electron transport rate, and ChlF emission, in response to water stress. We present the variations in the quantum yields of photochemical quenching in PSII (Ф_P_), fluorescence (Ф_F_), and regulated and basal heat dissipation (Ф_N_ and Ф_D_), namely, the probability of an absorbed photon being used in a given pathway, under different drought and light conditions. We pay particular attention to the response of the phase-shift in the Ф_F_-Ф_P_ relationship to water stress and its implications for interpreting the SIF-GPP relationship. By reformulating the mechanistic light–response (MLR) model [31], we are able to propose a SIF-based mechanistic model to accurately track rapid changes in plant photosynthesis status resulting from drought stress. We discuss (1) the mechanisms regulating the interaction among Ф_P_, Ф_F_, Ф_N_, and Ф_D_ under different degrees of water stress, (2) the reason why the drought response of satellite SIF observations is more pronounced than that obtained from leaf-level measurements, and (3) practical considerations regarding the application of the proposed model to large scales.

## 2. Results

### 2.1. The Cascade of Drought-Induced Changes in the Photosynthetic Parameters

The morphology of the plants in the WS treatment significantly changed in response to progressive drought stress over the time course of 1 to 28 days (WS1 to WS28), resulting in the curling (WS7 and 10), yellowing (WS14), drying (WS20) and, ultimately, the death of stems/leaves (WS26) (Figure 1). The *θ*_SWC_ remained at 19.1% in the control pots but declined in drought-treated pots as the drought intensified. The *θ*_SWC_ reduction rate was high during the early stages of drought, decreasing by nearly half from 19.1% on WS0 (no stress) to 9.2% on WS9 (Figure 1), with the *θ*_SWC_ decreasing at a slower rate to 7.3% on day 16 of the drought cycle (WS16) as the drought stress progressed (Figure 1). Subsequently, after the imposition of progressive drought for 18 days (WS18), the *θ*_SWC_ remained rather stable, only declining by 6.3% by WS28 (Figure 1). The variations in *β*_S_ and *β*_B_ as the soil dried are provided in the Appendix A.

Changes in the light–response curves of *A*_net_, G_S_, *J*_a_PAM_, and ChlF_P_F_ with progressive drought are shown in Figure 2. A strong drought response was observed in *A*_net_. Increasing water stress diminished both the maximum *A*_net_ value achieved and the irradiance at which this maximum was observed (Figure 2, first row). Light-saturated *A*_net_ decreased from 24.3 μmol m^−2^ s^−1^ at day 0 (prior to drought treatment) to 19.1 μmol m^−2^ s^−1^ at day 5, during which time SWC dropped from 19.1% to 12%. *A*_net_ decreased at a greater magnitude subsequently. *A*_net_ in the water-stressed plants fell to near-zero by day ten of the drought. Compared with *A*_net_, G_S_ showed an even more pronounced drought response (Figure 2, second row). Light-saturated G_S_ dropped by more than 60%, or from 0.54 mol m^−2^ s^−1^ to 0.21 mol m^−2^ s^−1^, after only 5 days of drought, and the stomata almost completely closed on the 8th day of treatment (i.e., G_S_ ≈ 0 during each light regime). Drought stress had a relatively smaller effect on the electron transport rate (Figure 2, third row). The maximum *J*_a_PAM_ reached 120.2 μmol m^−2^ s^−1^ after 5 days of water stress, or about 85% of the corresponding value at the beginning of the experiment. Over 10 days of drought, the maximum *J*_a_PAM_ still accounted for about 30% of that at WS0, while both *A*_net_ and G_S_ approached 0. The fluorescence response occurred much later, and was much smaller, than the responses of *A*_net_, G_S_, and *J*_a_PAM_ (Figure 2, bottom row); there were only small variations in the light–response curve of ChlF_P_F_ during the first 12 days into the drought. We found that absorbed PAR in the water-stressed plants remained almost unchanged during this 12-day period of water stress development (Appendix A), which may explain the weak drought response in the fluorescence emission. In fact, the ChlF_P_F_ value of the water-stressed plants maintained a fairly high level of ChlF_P_F_ after two weeks of water stress, and was still detectable at WS26 (Appendix A).

### 2.2. Drought-Induced Changes in the Photosynthesis–Fluorescence Relationship

We compared changes in ChlF_P_F_ during the light–response curves against the *A*_net_ of individual samples in the WS treatment under different water stress conditions: zero (*θ*_SWC_ = 19.1%), moderate (*θ*_SWC_ = 12.0%), and high (*θ*_SWC_ = 9.7%), which occurred at WS0, WS5, and WS8, respectively (Figure 3). Water stress regulated both the saturation levels of *A*_net_ and how the saturation level was approached as ChlF_P_F_ increased. Under no drought stress, *A*_net_ showed an initial linear increase with increasing ChlF_P_F_ and reached around 25 μmol m^−2^ s^−1^ when ChlF_P_F_ = 10 μmol m^−2^ s^−1^, after which it largely leveled off with a further increase in ChlF_P_F_ (Figure 3). As water stress increased, *A*_net_ tended to increase less steeply and reached a plateau at a lower ChlF_P_F_. For the plants exposed to moderate drought, for example, *A*_net_ started to remain stable at nearly 17 μmol m^−2^ s^−1^ when ChlF_P_F_ > 6 μmol m^−2^ s^−1^ (Figure 3). At severe water stress levels, the dynamic of ChlF_P_F_ was relatively less affected; ChlF_P_F_ still increased from 0 to 20 μmol m^−2^ s^−1^, with PAR ranging from 0 to 2100 μmol m^−2^ s^−1^. However, *A*_net_ started to saturate even when ChlF_P_F_ > 4 μmol m^−2^ s^−1^ and remained less than 8 μmol m^−2^ s^−1^ for the complete light–response curve (Figure 3).

### 2.3. The Variations in the Mechanisms Linking Photosynthesis and Fluorescence under Drought

The increasing nonlinearity in the relationship between *A*_net_ and ChlF_P_F_ (Figure 3) suggests that water stress causes shifts in the allocation of absorbed light energy dissipation pathways. At the beginning of the experiment (WS0, non-stress), Ф_P_ showed an inverse correlation with PAR and exhibited less sensitivity to PAR with increased light levels (Figure 4). The trade-offs between these yields appear to be governed by the complementarity between PQ and NPQ: Ф_N_ increased with increased PAR, and Ф_P_ + Ф_N_ ≈ 0.8 (Figure 4). In contrast, Ф_D_ and Ф_F_ had a muted sensitivity to changes in PAR: both of them showed a slight increasing trend at low light levels, and a decreasing trend at intermediate or high light levels (Figure 4).

During the first five days after imposing water treatment (WS1 to WS5), when *θ*_SWC_ decreased from 19.1% to 12.0%, the light–response curves of Ф_P_, Ф_N_, Ф_D_ and Ф_F_ showed a weak response to drought under low PAR conditions. For example, Ф_P_ taken at PAR = 180 μmol m^−2^ s^−1^ decreased slightly from 0.63 to 0.61. However, a further reduction in Ф_P_ occurred at higher PAR levels; for instance, Ф_P_ at PAR = 700 μmol m^−2^ s^−1^ decreased from 0.42 to 0.36 (Figure 4). Ф_N_ showed a small increasing trend during each light regime, with a larger magnitude under intermediate and high light conditions (Figure 4). Both Ф_D_ and Ф_F_ increased over the entire light–response curve (Figure 4).

From the 6th to the 8th day of treatment (WS6 to WS8), the variations in these four yields were still relatively limited when PAR ≤ 700 μmol m^−2^ s^−1^. However, Ф_P_ showed a significant decline at high PAR values: at WS8, Ф_P_ decreased to 0.13, 0.10, and 0.07 at 1300, 1700, and 2100 μmol m^−2^ s^−1^, respectively (Figure 4). NPQ still remained complementary to PQ: Ф_NPQ_ increased to 0.51, 0.62, 0.65 for these three PAR regimes (Figure 4). Their net effect on Ф_D_ and Ф_F_ was diminished; the light responses of Ф_D_ remained fairly unchanged and Ф_F_ showed a small decreasing trend (Figure 4).

During the period between 9 and 12 days (WS9 to WS12), after withholding water, when *θ*_SWC_ dropped below 9.0%, Ф_P_ decreased rapidly with increasing PAR, and this drop grew steeper as water stress developed (Figure 4). In other words, additional water stress increased the degree of nonlinearity in the relationship between Ф_P_ and PAR. We also found that a severe water deficit may reduce the complementarity between PQ and NPQ; Ф_N_ also showed a clear decreasing trend throughout the light–response curves (Figure 4). As these four pathways compete for absorbed energy, both Ф_D_ and Ф_F_ showed a clear increase: the maximum Ф_D_ and Ф_F_ reached 0.57 and 0.03, respectively, on the 12th day of treatment (Figure 4).

### 2.4. The Phase-Shift in the Relationship between Photochemical and Fluorescence Yields

The nonlinear relationship between Ф_F_ and Ф_P_ physiologically regulates the asymptotic behavior of the link between fluorescence and photosynthesis, that is, a positive or negative SIF-GPP relationship [32,33]. Similarly to Maguire et al. [34], we fitted a polynomial model to the relationship between Ф_F_ and Ф_P_, and the breakpoint was identified as the value of Ф_P_ where the slope of polynomial shifted from positive to negative (Figure 5). This breakpoint separates the relationship between Ф_F_ and Ф_P_ into two parts [3]: (1) Ф_F_ is proportional to Ф_P_ under low Ф_P_ (i.e., high light, ‘NPQ phase’), and (2) Ф_F_ is inversely proportional to Ф_P_ under high Ф_P_ (i.e., low light, ‘PQ phase’).

At WS0, the breakpoint was located at Ф_P_ = 0.46. Ф_F_ was positively correlated with Ф_P_ when Ф_P_ ≤ 0.46, and they were negatively correlated when Ф_P_ > 0.46 (Figure 5). The imposition of progressive drought for 5 days (WS1 to WS5) made the phase-shift in the Ф_P_-Ф_F_ relationship occur at a lower Ф_P_; the value of Ф_P_ at the breakpoints dropped from 0.46 to 0.37 (Figure 5). However, the breakpoints showed no clear trend during the period from WS6 to WS9 (Figure 5), most likely due to a decrease in Ф_F_ during that period (i.e., a flatter relationship between Ф_P_ and Ф_F_), and the limited number of light regimes in the middle of the light–response curves. The breakpoints were again observed to decline markedly after 10 days of progressive water stress. For example, the breakpoint at WS12 occurred at Ф_P_ = 0.12 (Figure 5).

### 2.5. The Performance of the rMLR Model

The trained parameters were applied to the testing dataset, and the resulting model performance in simulating *V*_cmax_, *J*_max_, Ф_P_, NPQ *A*_net_, and G_S_ was quantified using linear regression analysis and described using the coefficient of determination (*R*^2^) and the root mean squared error (RMSE) between the simulated and measured values.

A relatively small decrease was observed in both *V*_cmax_ and *J*_max_ between WS1 and WS6 (Figure 6). As the drought continued, however, they dropped substantially, and were almost zero after WS10 (Figure 6). As water stress progressed, the rMLR model was able to track the decreasing trends in *V*_cmax_, and *J*_max_ well, explaining 83% (RMSE = 8.25 μmol m^−2^ s^−1^, Figure 6a) and 79% (RMSE = 24.35%, Figure 6b) in their variance between WS1 and WS12. However, the model tended to underestimate the large values of *V*_cmax_, and *J*_max_ during WS1-WS6, and overestimate the small values between WS7 and WS9 (Figure 6).

During the first week of the experiment, Ф_P_ gradually declined from 1.0 to approximately 0.3 as PAR increased from 0 to 1000 μmol m^−2^ s^−1^ (Figure 7). As the drought intensified, Ф_P_ became less responsive to PAR: Ф_P_ decreased more steeply with increased PAR and leveled off over a broader range of light intensities (Figure 7). The simulated Ф_P_ light-curve shapes were similar to those estimated from the fluorescence parameters: between WS1 and WS12, the model explained 94.3% of the variation in Ф_P_ (RMSE = 0.07, Figure 7). Under severe water stress conditions (WS8–WS12), the model underestimated Ф_P_, with *R^2^* = 0.93 (RMSE = 0.08, Figure 7).

The NPQ light–response curves have an approximately opposite form to those of Ф_P_. NPQ tended to saturate at lower PAR levels with an increase in the water deficit, and substantially decrease under severe stress (Figure 8). The proposed model was able to track the light response of NPQ well: *R^2^* between measured and modelled NPQ reached a value of 0.97 (RMSE = 0.16) for the period between WS0 and WS12 (Figure 8). The decrease in *θ*_SWC_ had no obvious effect on the performance of the model in simulating NPQ, with *R*^2^ remaining at 0.97 between WS9 and WS12 (RMSE = 0.16, Figure 8). Despite the overall good performance, a small underestimation in NPQ was observed under high light intensities (PAR > 1700 μmol m^−2^ s^−1^, Figure 8).

The simulated response of *A*_net_ to variations in light intensity shows a similar pattern to the response measured by the gas-exchange system: simulated *A*_net_ increases rapidly with increasing illumination intensity at low light levels, and gradually reaches a plateau under high light conditions (Figure 9). The rMLR model is able to reproduce *A*_net_ well under drought conditions; it accounted for 97.2% (RMSE = 1.532 μmol m^−2^ s^−1^) of the variability in *A*_net_ from WS0 to WS12 (Figure 9). However, the model did tend to consistently overestimate *A*_net_, with the degree of overestimation appearing to be independent of the severity of drought; simulated *A*_net_ was about 15% higher than measured *A*_net_ in non-stressed plants and in plants subjected to mild drought (Figure 9), with the extent of overestimation varying between 10% and nearly 50% for moderate and severe drought, respectively (Figure 9). The constant value used for the Ф_Pmax_ value (0.8) is one possible explanation for the overestimation. A decreased Ф_Pmax_ has been suggested to occur under water stress [1,35,36]. An overestimation of Ф_Pmax_ in drought would lead to an overestimation in χ (Equation (12)) and then in NPQ (Equation (10)), and would consequently lead to a higher simulated *A*_net_ than that observed (NPQ occurs in the numerator of the rMLR model, Equation (3)).

Overall, the simulated G_S_ can explain 91.7% (RMSE = 0.044 μmol m^−2^ s^−1^) of the variability in the measurements of G_S_ collected between WS0 and WS12 (Figure 10). The rMLR model overestimated G_S_ for non-stressed plants (WS0) by about 15%. However, the overestimation was suppressed from the beginning of the drought-stress period: the RMSE between simulated and measured G_S_ decreased to 0.040 μmol m^−2^ s^−1^ (WS1–WS12) for the water-stressed plants (Figure 10). In particular, the proposed model demonstrated good potential for mimicking the fast drop in stomatal conductance (*R^2^* = 0.89 and RMSE = 0.035 μmol m^−2^ s^−1^) in the early stages of water stress (WS1–WS4). The overestimation of G_S_ at WS0 can be explained by the overestimation in *A*_net_ (Figure 9). The decrease in *β*_S_ as drought became more severe tended to cancel out the negative effect of overestimation in *A*_net_ (Equation (21)), resulting in an improvement in the simulation of G_S_.

## 3. Discussion

Identification of the cascade of drought-induced changes in photosynthetic characteristics is highly relevant to fully understanding the relationships between photosynthesis, NPQ, and fluorescence under water stress conditions. Stomatal closure is the earliest response to drought, paralleling a decrease in photosynthetic CO_2_ assimilation but with a lower magnitude (Figure 2). In the early stages of drought stress, exposure to the volatile plant hormone methyl jasmonate (MeJA) can mitigate drought stress by stomatal closure [37,38,39] During WS1–WS6, under mild-to-moderate water stress, energy partitioning in PSII (i.e., Ф_P_, Ф_N_, Ф_D_, and Ф_F_) remained virtually unchanged under low or intermediate light levels (Figure 4). However, we noticed a decline in Ф_P_, and a rise in the other three pathways, Ф_D_, Ф_N_ and Ф_F_, in the presence of high light levels during that period (Figure 4). These observations may suggest that the proportion of energy dissipated in each pathway does not change much regardless of stomatal closure levels during the initial period of drought [40].

At moderate-to-severe drought (WS7–WS9), G_S_ almost vanished (Figure 2) and the observed drop in *A*_net_ indicated the predominance of non-stomatal limitations to photosynthesis [41]. Given the lower G_S_, the excess light energy can ultimately lead to the production of reactive oxygen species and reflect the MeJA response [37,38,39]. As a consequence, Ф_P_, the quantum yield of photochemistry in PSII, showed a decreasing trend, especially at high light levels (Figure 4). To compensate for the decline in Ф_P_, regulated heat dissipation (Ф_N_) tended to increase as a main photoprotective mechanism, and Ф_D_ remained fairly constant (Figure 4). The engagement of thermal energy dissipation also resulted in a small decrease in Ф_F_.

Under extreme drought conditions (WS10–WS12), Ф_P_ continued to decrease, and the stressed plants kept increasing their use of thermal energy dissipation and fluorescence emission (Ф_F_) to cope with excess light energy (Figure 4). However, the process of thermal energy dissipation changed from a regulated form that can be rapidly activated by excess light to a sustained form that is rather insensitive to fluctuating light [42,43], resulting in the simultaneous rise of Ф_D_ and decline of Ф_N_ (Figure 4). This result is consistent with the findings that this transformation of energy dissipation characteristics tends to occur during periods of harsh environmental stress [44].

The results confirm that water stress increases nonlinearity in the overall relationship between photosynthesis and fluorescence (Figure 3). Under typical high light levels, as what might occur for early afternoon spaceborne SIF retrievals, the ‘NPQ phase’, driving a positive SIF-GPP relationship, is representative of non-stressed plants. We showed that the transition from the ‘PQ phase’ to the ‘NPQ phase’ is driven by both irradiance and the severity of drought. During moderate or severe drought stress, the decrease in Ф_P_, and increase in Ф_F_, pushes the stressed plant from the ‘NPQ phase’ to the ‘PQ phase’, resulting in a more nonlinear SIF-GPP relationship when considering both the drought and non-drought periods together.

When the stomata are closed, Marrs et al. (2020) [16] found that the photosynthetic rate decreases rapidly, and SIF is relatively less affected. SIF cannot track the changes of photosynthesis, resulting in the phenomenon of decoupling fluorescence and photosynthesis. During a short drought duration, Helm et al. (2020) [29] showed that the degree of reduction in leaf photosynthesis was much greater than the reduction in SIF. These studies are consistent with our findings. Our results also show that the fluorescence response to water stress occurs later, and at a smaller magnitude, which appears to contradict studies using satellite SIF, suggesting a strong negative response [26,27]. Such a dichotomy can be resolved by assessing the relative importance of the structural and physiological contributions to the drought response of fluorescence at different scales. Fluorescence variations reported in leaf-scale studies are driven by changing fluorescence efficiencies alone (unless absorbed PAR is unchanged) which, as shown here, has a relatively muted sensitivity to water stress. However, at the whole-plant or canopy scales, both structural and physiologic components may regulate TOC SIF, and so a drought-induced decline in absorbed PAR due to canopy structural changes (i.e., changes in leaf area or leaf angles) dominates negative anomalies observed in satellite SIF [27,45]. It is worth noting that these seemingly inconsistent results should not necessarily be viewed as a failure of SIF for monitoring plant water stress, as the changes in canopy structure may also directly reflect the plant water status. The inconsistency also highlights the need to consider/normalize canopy structure factors during drought. Otherwise, we may run the risk of wrongly assigning physiological causality to variance in TOC SIF due to changes in the canopy structure [28].

Equation (3) also contains other parameters (for example, *K*_DF_, *K*_mc_, *K*_mc_, *Г** and Φ_Pmax_). These additional parameters are often assumed to be constant in the literature but may actually vary with water stress. The actual *K*_DF_ value is currently unknown. Gu et al. (2019) [31] set *K*_DF_ to 19, while Liu et al. (2022) [46] assumed *K*_DF_ to be 9. According to Equation (3), using these two *K*_DF_ values will directly cause *A*_net_ to change by a factor of two. *K*_DF_ remains unchanged at the same temperature, but there is currently no relevant research on *K*_DF_ under water stress. *K*_mc_, *K*_mc_ and *Г** depend on the partial pressure of oxygen and temperature, which are related to the specificity factor of Rubisco [47]. Furthermore, Φ_Pmax_ decreases under water stress conditions. Considering these potential effects of water stress on the parameters involved in our theoretical equations, more future measurements are needed to quantitatively examine their relationships under water conditions. We performed a sensitivity analysis on the MLR model. The input variables are *C*_i_, *K*_co_, *Γ**, Φ_Pmax_, *q*_L_, and ChlF_P_F_, respectively. The sensitivity analysis results show that the main parameters affecting *A*_net_ variation are leaf physiological parameters, *q*_L_ and ChlF_P_F_. These two parameters explain more than 80% of the *A*_net_ variation, while other leaf physiological parameters with greater influence such as *C*_i_, *K*_co_, and Φ_Pmax_ account for more than 15% of the total variation. However, the parameter *Γ** plays a small role in explaining the *A*_net_ variation. Han et al. (2022) [47] studied the relationship between SIF and GPP through the MLR model and showed that the *q*_L_ is sensitive to light intensity and can be expressed by the exponential equation of two parameters (a*q*_L_ and b*q*_L_) between the *q*_L_ and PAR. However, we found that in the uniform plant functional types, the parameters a*q*_L_ and b*q*_L_ vary greatly. Therefore, we use the NPQ/Φ_P_ version of the *J*_a_-ChlF_P_F_ equation. The NPQ/Φ_P_ version mixes photochemistry and non-photochemistry. Further, this mixing is superficial because information on non-photochemistry is canceled out in the product of (1 + NPQ) and Φ_P_/(1 − Φ_P_) as Φ_P_ contains both photochemical and non-photochemical information. We need to consider both the energy-dependent and energy-independent components of NPQ. This is particularly important for this study because it focuses on water stress, which likely induces energy-independent NPQ.

The weak response of fluorescence to a water deficit suggests that SIF alone, at least its physiological component, is not able to track drought-induced changes in plant physiology. However, the development of the rMLR model allows for a mechanistic understanding of the drought impact on photosynthetic characteristics using SIF, soil moisture, and two measurable meteorological variables as the inputs. To apply the rMLR model at the canopy scale, that is, by using narrowband SIF*_toc_* as an input variable for Equation (3), several more steps will need to be performed. First, the contribution of PSII fluorescence to SIF*_toc_* (*f*_PSII_) should be determined. Although Bacour et al. [48] showed that *f*_PSII_ can be also estimated from *T_air_* and PAR, more research is needed to assess how *f*_PSII_ varies with species and environmental conditions [44]. Second, the probability of a fluorescence photon escaping from a leaf level to canopy scale (*f*_esc_L-C_) must be quantified, in addition to *f*_esc_P-L_ (Equation (8)). In the near-infrared (NIR) region, *f*_esc_L-C_ can be estimated from directional reflectance (RNIR) [49]. Note that both SIF*_toc_* and *R*_NIR_ can be concurrently obtained from measurements of irradiance/radiance [50]. Thus, the estimation of *f*_esc_L-C_ requires no additional observations. Third, a full-band SIF emission should be reconstructed from narrowband SIF. The full SIF spectrum at TOC can be approximated by a linear combination of basis spectra [46,51,52]. The Soil-Canopy Observations of Photosynthesis and Energy Balance (SCOPE) [53] model is typically used to generate a dataset that is representative of the majority of actual scenes. Principal Component Analysis (PCA) or Singular Value Decomposition (SVD) techniques are then applied to extract the basis spectra from this simulated dataset. However, the SCOPE model is designed for homogeneous vegetation canopies, such as crops, and its performance may deteriorate for a heterogeneous, structurally complex canopy [54]. Finally, a large-scale, and near-real-time root-zone soil moisture (RZSM) dataset is needed to estimate *β*_S_ and *β*_B_. Since satellite soil moisture satellite observations are sensitive to surface soil moisture, typically within the first few centimeters, current RZSM datasets are mostly produced by assimilating observations into model simulations [55,56].

## 4. Materials and Methods

### 4.1. Leaf-Scale Concurrent Instrumentation

We developed a leaf-scale concurrent measurement system by integrating a portable gas-exchange system (LI-COR Biosciences, Lincoln, NE, USA), two HR2000+ spectrometers and two QE *Pro* spectrometers (Ocean Optics, Dunedin, FL, USA), a PAM system (Dual-PAM-100, Heinz Walz GmbH, Effeltrich, Germany), a short-pass filter, an external LED light source and fiber optics (connecting the PAM and spectrometers to the leaf chamber) (Figure 11). With this measurement system, we were able to simultaneously measure gas-exchange, passive and active ChlF, reflectance and transmittance for plants under a variety of controlled environmental conditions [46,57]. The main components and modifications are discussed below.

#### 4.1.1. Gas-Exchange System

We used the LI-6800 gas-exchange system to control the environmental conditions of a modified leaf chamber. The original film of the LI-6800 transparent leaf chamber was replaced with high-transparency plexiglass. An external LED light source (see below) was fixed to the high-transparency plexiglass via a cylindrical plastic tube. The optical fibers of the PAM fluorometer and spectrometer were inserted into the leaf chamber via two fiber adapter bulkheads added to the plastic tube. These optical fibers were fixed to the diagonal of the fiber adapter bulkhead. In order to measure downward chlorophyll fluorescence spectral radiant energy fluxes, a metal plate covering the base of the leaf chamber was first added, and a fiber adapter bulkhead for inserting optical fibers was then fixed in the middle of this plate. All the fiber adapter bulkheads were sealed with clay to ensure the leaf chamber remained air-tight. To reduce light scattering in the leaf chamber, the inside of the light-source plastic tube and the upper surface of the metal plate connected to the base of the leaf chamber were painted with black acrylic paint (Black 2.0, Stuart Semple, UK) as a light trap (Figure 11b). The modified gas-exchange system blocks all light, except for incoming light with wavelengths longer than 625 nm, using a short-pass filter.

#### 4.1.2. Gas-Exchange System

A Dual-PAM-100 instrument (Heinz Walz GmbH, Effeltrich, Germany) was used to measure fluorescence parameters. The Dual-PAM-100 featured a single red (625 nm) power LED for excitation of chlorophyll fluorescence. The saturating pulse was also driven by the red LED, which emitted a consistent 10,000 μmol m^−2^ s^−1^ for 0.8 s. The Dual-PAM-100 induced the maximum fluorescence quantum yield using the saturation pulse under dark-adapted conditions. The PAM was connected to the leaf chamber via a 1 m long single fiber of 0.8 cm diameter routed through the entrance hole (Figure 11a). The angle between the PAM fiber tip inserted into the entrance hole and the plane of the leaf chamber was kept at about 90° and the vertical distance between the fiber optic head and the surface of the leaf sample was maintained at 1 cm to avoid blocking the field of view of the PAM instrument.

#### 4.1.3. Spectrometers

Using two HR2000+ and two QE *Pro* (Ocean Optics, Dunedin, FL, USA) high-sensitivity spectrometers, we simultaneously measured reflection, transmission, and upward and downward chlorophyll fluorescence spectral radiant energy fluxes of the leaf (Figure 11c). The leaf chamber was linked to the four spectrometers by means of two bifurcated optical fibers. One of these fibers was inserted into the top fiber adapter bulkhead above the leaf chamber to connect to an HR and a QE *Pro*, while the other was inserted into the bottom of the leaf chamber and connected to the other HR and QE *Pro.* Both bifurcated fibers had a diameter of 1000 um to ensure they could collect enough light. The vertical distance between the tip of the top fiber and the sample surface of the leaf was kept at 10 mm to allow for measurement of the upward chlorophyll fluorescence. The lower fiber optic head was inserted vertically into the base of the leaf chamber to measure the downward chlorophyll fluorescence, with the distance from the back of the leaf being 10 mm. The HR2000+ spectrometers covered the 296–1203 nm range at an optical resolution of 5.3159 nm with a spectral sampling of 0.4430 nm. The detector of the QE *Pro* spectrometer covered the range 634–863 nm at an optical resolution of 5.2656 nm with a spectral sampling of 0.2194 nm. The absolute calibration of the spectrometers and all light paths were carried out by using a separate reference QE *Pro* spectrometer (Ocean Optics, Dunedin, FL, USA) calibrated using a NIST traceable integration sphere. Dark current and non-linear spectrometer calibrations were completed before each measurement.

#### 4.1.4. Light Source

The external actinic light source was a white LED light source (S5000, Nanjing Hecho Technology Co., Nanjing, China) with a ring-shaped fiber which provided a homogeneous light distribution across the leaf chamber. The actinic light source was capable of delivering 0–3000 μmol m^−2^ s^−1^ with a 400–700 nm wavelength range. To attenuate this LED light source, a customized 625 nm short-pass, fused silica filter with a diameter of 12.5 mm was placed between it and the O-ring fiber optic (Figure 11). After short-pass filtering, the rejection wavelength range of the external light source was, in fact, 639–925 nm, due to insufficient accuracy. In the 350–612 nm range, the transmittance exceeded 91% and the optical density was 4 (Edmunds Optics, Barrington, NJ, USA). A tin-foil lampshade outside the cylindrical plastic tube was used to block outside light. The light intensity was varied in the range of 0–2500 μmol m^−2^ s^−1^ by adjusting the pulse-width modulation controller in the light source.

### 4.2. Experiment Design

The experiment was conducted between 15 October 2020 and 30 April 2021 in the Northwest A&F University, Yangling, China. Winter wheat (*Triticum aestivum* L. cultivar Xi Nong 979) was sown in plastic pots (27 cm height × 21 cm diameter) filled with 9 kg of sieved, air-dried, loess topsoil and 1.95 g of urea. The field capacity of the soil was 24% and the wilting point was 9%. The plants were grown under a rainproof shed until the turning-green stage, and then moved to a climate chamber where CO_2_ concentration, air temperature and relative humidity were controlled at 400 µmol mol^−1^, 12 °C and 60%. The light intensity above the canopy during the experiment was kept at 600 μmol m^−2^ s^−1^ for 12 h per day, from 8 a.m. to 8 p.m. Beginning in late March, eight representative wheat plants in the heading stage were chosen for the experiment.

Throughout the 28-day experiment, the gravimetric soil water content (*θ*_SWC_, %) was continuously monitored using the weighing method [58,59]. The wheat plants were subjected to one of two treatments: non-water stress (NS) and water stress (WS). Four wheat plants in each treatment were randomly selected as biological replicates. In the NS treatment, the *θ*_SWC_ was maintained at 19.1% (well-watered plants, assuming 80% of field capacity) [60,61] throughout the experiment, but, in the WS treatment, to simulate intensifying drought, *θ*_SWC_ was gradually decreased to a final value of 6.3%. This steady decline in *θ*_SWC_ ensured that plants under the WS treatment were subjected to progressive drought stress [62].

Measurements were made on attached leaves of the wheat plants under the two water treatments. The wheat leaf had to be placed in the 3 cm× 3 cm clear chamber along the diagonal of the square and positioned in the chamber with its adaxial surface facing the LED light source. Since the leaves did not fill the leaf chamber, the area of each leaf had to be accurately measured. After 1 h of dark adaptation, a saturation flash from the PAM fluorometer was used to determine minimal fluorescence (F_o_) and maximal fluorescence in the dark (F_m_) of dark-adapted leaves. Next, the light–response curves and CO_2_ response curves of gas exchange and fluorescence were measured. The CO_2_ flow rate, air relative humidity and leaf temperature were kept constant at 500 μmol s^−1^, 50% and 12 °C, respectively. To obtain light–response curves, measurements were conducted at a CO_2_ concentration of 400 µmol mol^−1^. Light–response measurements were made with photochemically active radiation (PAR) light intensities of 0, 40, 90, 180, 350, 700, 1300, 1700, and 2100 µmol m^−2^ s^−1^. Next, we measured the CO_2_ response curve under saturated light intensity (1500 µmol m^−2^ s^−1^) using values of the CO_2_ concentration gradient of 30, 50, 100, 200, 300, 400, 600, 900, 1200, 1500 μmol mol^−1^.

Throughout the course of each light–response curve and CO_2_ response curve determination, spectral measurements (reflected radiance, transmitted radiance, forward and backward fluorescence spectral radiant energy flux) were continually recorded at 1 s intervals. Steady-state fluorescence emission (F_t_), induced by the measuring beam of the PAM fluorometer, was also included. Maximal fluorescence emission in the light-adapted state (F_m_’) from the PAM was recorded under each light intensity and CO_2_ concentration. From these measurements (F_o_, F_t_, F_m_, F_m_’), we also obtained NPQ, a fraction of open PSII reaction centers (*q*_L_) [63], and the actual rate of electron transport (*J*_a_PAM_, μmol m^−2^ s^−1^). The details for estimating them are provided in Appendix A. Net CO_2_ assimilation rate (*A*_net,_ μmol m^−2^ s^−1^) and stomatal conductance to water vapor (G_S_, mol m^−2^ s^−1^) provided by the gas-exchange system were automatically stored every 5 s. To ensure that the gas-exchange conditions were stable, the measurements for light curves and CO_2_ curves were made after waiting at least 5 min, and up to 20 min, between each light intensity or CO_2_ concentration change. The filtered incident radiation of the LED light source at each light intensity were measured with a standard reflectance panel (Spectralon; Labsphere, North Sutton, NH, USA). The above measurements were taken at day 0 (before the water stress treatment started, WS0) and at days 2, 4, 5, 6, 7, 8, 9, 10, 12, 14, 16, 18, 20, 22, 24, 26, 28 after withholding water (WS2, WS4, etc., respectively). At WS0, the unstressed maximum carboxylation capacity of Rubisco (*V*_cmax,0_, μmol m^−2^ s^−1^) and the unstressed maximum electron transport rate (*J*_max,0_, μmol m^−2^ s^−1^) were estimated by fitting the FvCB model [64] to the CO_2_ response curves. According to the photosynthetic light–response curve of winter wheat at WS0, *A*_net_ increased steeply with PAR when PAR ≤ 350 μmol m^−2^ s^−1^, and increased slowly with increasing PAR when PAR ≥ 1000 μmol m^−2^ s^−1^ (see below). Thus, a PAR level between 350 and 1000 μmol m^−2^ s^−1^ was assumed to be the intermediate light condition, and PAR levels lower than 350 or higher than 1000 μmol m^−2^ s^−1^ were taken to be low and high light conditions, respectively.

### 4.3. The Reformulated MLR (rMLR) Model

The MLR model [31] shows that *J*_a_ can be mechanistically estimated from *q*_L_, Ф_Pmax_, and the chlorophyll fluorescence flux density emitted from the photosystem II (PSII) across the full ChlF emission spectrum (ChlF_P_F_, μmol m^−2^ s^−1^). At the photosystem level,
(1)$$J_{a}=\frac{q_{L}\times \Phi_{{}_{\mathrm{Pmax}}}\times (1+K_{{}_{\mathrm{DF}}})\times {\mathrm{ChlF}}_{P\_F}}{\left(1-\Phi_{{}_{\mathrm{Pmax}}}\right)},$$
where *K*_DF_ is the ratio between the rate constants for constitutive heat loss (*K*_D_) and fluorescence (*K*_F_) and is assumed to be 9 [46]. However, *q*_L_ is much less studied than the other PAM parameters [31]. A previous study [46] shows that the role of *q*_L_ and Ф_Pmax_ in the original MLR model can be replaced by Ф_P_ and NPQ:(2)$$J_{a}=\frac{\Phi_{{}_{P}}\times (1+\mathrm{NPQ})\times (1+K_{{}_{\mathrm{DF}}})\times {\mathrm{ChlF}}_{P\_F}}{\left(1-\Phi_{{}_{P}}\right)},$$

The practical advantage of Equation (2) is that both Ф_P_ and NPQ (dimensionless) can be estimated from air temperature (*T*_air_, °C) and PAR (see the section on the estimation of Ф_P_ and NPQ). Note that *J*_a_ in Equations (1) and (2) is already balanced by carboxylation and photorespiration [31,47]. To differentiate from *J*_a_PAM_, in this study, *J*_a_ was calculated from ChlF emission using Equation (2).

One can obtain *A*_net_ for C_3_ and C_4_ species:(3)$$A_{\mathrm{net}}=A_{g}-R_{d}=\left\{ \begin{array}{l} \frac{C_{c}-\Gamma *}{4C_{c}+8\Gamma *}\times \frac{\Phi_{{}_{P}}\times (1+\mathrm{NPQ})\times (1+K_{{}_{\mathrm{DF}}})\times {\mathrm{ChlF}}_{P\_F}}{\left(1-\Phi_{{}_{P}}\right)}-R_{d}\;C_{3}\;\\ \frac{1-\zeta }{3}\times \frac{\Phi_{{}_{P}}\times (1+\mathrm{NPQ})\times (1+K_{{}_{\mathrm{DF}}})\times {\mathrm{ChlF}}_{P\_F}}{\left(1-\Phi_{{}_{P}}\right)}-R_{d}\;\;\;C_{4}\; \end{array}\right.,$$
where *A*_g_ represents gross photosynthesis (μmol m^−2^ s^−1^); *R*_d_ is the daytime respiration (μmol m^−2^ s^−1^); *C*_c_ is the chloroplastic CO_2_ partial pressure (µmol mol^−1^); *Γ** is the chloroplastic compensation point of CO_2_ µmol mol^−1^ [31,65]; *ζ* is the fraction of total electron transport of mesophyll and bundle sheath allocated to mesophyll, assumed to be 0.4 [66]. We refer to Equation (3) as the rMLR model. The procedure for quantifying *A*_net_ from the observed SIF is illustrated in Figure 12.

### 4.4. Correction for the PSI Fluorescence

The rMLR model is only valid for fluorescence emissions from PSII [31]; the contribution of the PSI fluorescence should be excluded from all actively and passively induced ChlF related to Equation (3). Using the far-red method, Pfündel et al. [67] showed that the PSI fluorescence yield (F_1_) detected by the PAM fluorometer for C_3_ species can be estimated as:(4)$$F_{{}_{1}}=0.24\times F_{{}_{o}},$$

By subtracting F_1_ from the fluorescence yields (F_o_, F_m_, F_m_′, F_t_) directly measured by the PAM fluorometer, we were able to correct them for PSI fluorescence. Accordingly, all other yields/parameters derived from these four yields, Ф_P_, Ф_F_, Ф_N_, Ф_D_, and NPQ, were also corrected for PSI fluorescence. See the details in Appendix A. Hereafter, all of the active fluorescence parameters in this study only contain the PSII contribution unless otherwise specified.

The leaf-scale concurrent measurement system (Figure 11) provides the passive ChlF spectrum in the range 640 to 850 nm at the leaf scale (ChlF_L_(λ), mW m^−2^ nm^−1^ sr^−1^):(5)$${\mathrm{ChlF}}_{{}_{L}}(\lambda ){=\mathrm{ChlF}}_{{}_{L\_U}}(\lambda )+{\mathrm{ChlF}}_{{}_{L\_D}}(\lambda ),$$
where ChlF_L_U_(λ) and ChlF_L_D_(λ) represent the fluorescence radiance emitted from adaxial and abaxial leaf surfaces (μW cm^−2^ nm^−1^ sr^−1^), respectively, and *λ* is the wavelength (nm).

Due to the strong linear relationships between spectral fluorescence yields and the original PAM F_t_ yields [68,69], deriving the ratio of F_t_ yields before and after correction for PSI fluorescence can also allow for an approximate separation of the PSI and PSII spectral fluorescence yields. Further, Pfündel [70] showed that the ratio of F_1_ to F_o_ was 14% and 45% in the spectral ranges below 700 nm (SW) and above 700 nm (LW), respectively. Considering the wavelength-dependent relationships among spectral and PAM fluorescence yields [69], the contribution of PSII (ChlF_L_PSII_(λ)) to measurements of ChlF_L_(λ) can be estimated as:(6)$${\mathrm{ChlF}}_{{}_{L\_\mathrm{PSII}}}\left(\lambda \right)=\left\{ \begin{array}{l} {\mathrm{ChlF}}_{{}_{L}}\left(\lambda \right)\times \frac{F_{{}_{t}}-F_{{}_{1\_\mathrm{SW}}}}{F_{{}_{t}}}\;\;\lambda \leq 700\;\mathrm{nm}\\ {\mathrm{ChlF}}_{{}_{L}}\left(\lambda \right)\times \frac{F_{{}_{t}}-F_{{}_{1\_\mathrm{LW}}}}{F_{{}_{t}}}\;\;\lambda >700\;\mathrm{nm} \end{array}\right.,$$
where F_t_ is the steady-state fluorescence yield, and F_1_SW_ (F_1_SW_ = 0.14 × F_o_) and F_1_LW_ (F_1_LW_ = 0.45 × F_o_) represent the PSI contribution at SW wavelengths and LW wavelengths, respectively. Note that F_t_ and F_o_ used in Equation (6) were directly measured by the PAM fluorometer and thus contain contributions from both PSI and PSII.

To apply the rMLR model, ChlF_L_PSII_(λ) must be further downscaled to the photosystem level (ChlF_P_(λ)) by accounting for the probability that a fluorescence photon escapes from the PSII light reactions inside the leaves to the surface of the leaf (*f*_esc_P-L_):(7)$${\mathrm{ChlF}}_{P}(\lambda ){=(\mathrm{ChlF}}_{{}_{L\_\mathrm{PSII}}}(\lambda ))/f_{{}_{esc\_P-L}}(\lambda ),$$

*f*_esc_P-L_ is approximately equal to the sum of leaf reflectance (*R*) and transmittance (*T*) [71]:(8)$$f_{{}_{\mathrm{esc}\_P-L}}\approx R(\lambda )+T(\lambda )\;,$$

Note that ChlF_P_(λ) in Equation (7) has units of mW m^−2^ nm^−1^ sr^−1^. To obtain ChlF_P_F_ as required by Equation (3), we need to integrate ChlF_P_(λ) between 640 and 850 nm and perform a unit conversion:(9)$${\mathrm{ChlF}}_{P\_F}=\sum_{\lambda =640}^{850}\left[{\mathrm{ChlF}}_{P}(\lambda )\right]\times \frac{\lambda \times 10^{6}}{h\times c\times N_{A}\times 10^{3}\times 10^{9}}\;,$$
where *h* is the Planck constant (6.62607015 × 10^−34^ J·s), *c* is the light speed (3 × 10^8^ m s^−1^), *N*_A_ is the Avogadro constant (6.02 × 10^23^ mol^−1^), 10^6^ is used to convert moles (mol) to micromoles (μmol) in *N*_A_, 10^3^ is used to convert milliwatts (mW) to Watts (W), and 10^9^ is used to convert nanometers (nm) to meters (m) in *λ*. For the application of the rMLR model at the canopy scale or beyond, see the see the Section 3.

### 4.5. Estimation of *Ф_P_* and NPQ

In this study, *K*_D_ and *K*_F_ are assumed to be 0.9 and 0.1, respectively [46]. Note that NPQ should be equal to the rate coefficients of energy-dependent heat dissipation (*K*_N_) because NPQ = *K*_N_/(*K*_F_ + *K*_D_), and *K*_F_ + *K*_D_ = 1. *K*_N_ can be estimated as [48]:(10)$$K_{{}_{N}}=a\times \chi^{c}\times \frac{1+b}{b+\chi^{c}}\times \frac{\exp^{(d\times T_{\mathrm{air}}+e)}}{{\mathrm{PAR}}^{f}},$$
where a, b, c, d, e and f are fitting parameters. Following Bacour, Maignan, MacBean, Porcar-Castell, Flexas, Frankenberg, Peylin, Chevallier, Vuichard and Bastrikov [48], b, c, d, e and f are assumed to have values of 5.74, 2.167, −0.014, −0.00437 and 0.00576, respectively. Potentially, we can use an exponential equation with three parameters to represent the relationship between the parameter a and *θ*_SWC_:(11)$$a=g\times \exp^{((-h)\times \theta_{\mathrm{swc}})}+j,$$
where g, h, and j are empirical parameters. The measurements are randomly divided into two groups, with 50% of the data used for training, and the remaining 50% for evaluating the performance of the predictions. The Matlab function ‘lsqnonlin’ was used to obtain the values of the parameters by minimizing a cost function on the training dataset: *C* = (*M − S*)^2^, where *M* is the NPQ measured with the PAM, and *S* is the modelled NPQ. A Trust Region algorithm was used to update the values of the parameters after each iteration step, with the iteration terminating when the improvement in the cost function was less than 10^−3^. Appendix A presents the initial values, boundaries and constraints of the parameters.

*χ* is defined as [32,46]:(12)$$\chi =1-\frac{\Phi_{{}_{P}}}{\Phi_{{}_{\mathrm{Pmax}}}},$$
where Ф_Pmax_ is assumed to be 0.8 and is similar among healthy plants; Ф_P_ is estimated as [32]:(13)$$\Phi_{P}=\frac{{\min(A}_{{}_{C}}{,A}_{{}_{J}})}{\alpha_{\mathrm{grn}}\times \beta_{\mathrm{PSII}}\times \mathrm{PAR}}\times \frac{4C_{c}+8\Gamma *}{C_{c}-\Gamma *},$$
where A_C_ and A_J_ represent Rubisco-limited and RuBP-limited gross CO_2_ assimilations (μmol m^−2^ s^−1^), respectively. α_grn_ represents the absorption efficiency of PAR by green leaves, and the value is usually fixed at 0.84. *β*_PSІІ_ is the fraction of absorbed energy allocated to PSII, and the value is set to 0.5 [72]. A_C_ and A_J_ are given by [73]:(14a)$$A_{{}_{C}}=\frac{V_{{}_{\mathrm{cmax}}}\times (C_{c}-\Gamma *)}{C_{c}+K_{{}_{\mathrm{mC}}}\times (1+O_{{}_{c}}/K_{{}_{\mathrm{mO}}})},$$
(14b)$$A_{{}_{J}}=\frac{J_{{}_{p}}}{4}\frac{\left(C_{c}-\Gamma *\right)}{C_{c}+2\Gamma *}$$
where *K*_mC_ is the Michaelis–Menten constants of Rubisco for CO_2_ 270 µbar, [73]; *K*_mO_ is the Michaelis–Menten constants of Rubisco for O_2_ 16,500 µbar, [73]; *O*_c_ is the chloroplastic O_2_ partial pressure, assumed to equal to the oxygen partial pressure 230,000 µbar, [74]; *J*_p_ is the potential electron transport rate [47]:(15)$$J_{{}_{p}}=\frac{\sigma \times \mathrm{PAR}+J_{{}_{\max}}-\sqrt{{\left(\sigma \times \mathrm{PAR}+J_{{}_{\max}}\right)}^{2}-4\times \theta \times \sigma \times \mathrm{PAR}\times J_{{}_{\max}}}}{2\times \theta },$$
where σ is the product of leaf light absorptance, a fraction of absorbed photons allocated to PSII and Ф_Pmax_. σ is set to 0.3 [72]. *θ* is an empirical curvature parameter, which is also modelled as a function of *θ*_SWC_:(16)$$\theta =k\times \exp^{\left((-l\right)\times \theta_{\mathrm{swc}})}+m,$$
where k, l, and m are the fitting parameters. Again, 50% of the measurements were used to determine the parameter values by minimizing the squared difference between the measured and simulated values of Ф_P_ (Appendix A).

*A*_net_ is limited by biochemical processes under water stress, such that a soil-moisture-dependent stress function (*β*_B_, see Equation (22) below) should be applied to regulate the parameters *J*_max_ and *V*_cmax_ of the photosynthesis model [75]:(17a)$$J_{{}_{\max}}=\beta_{{}_{B}}\times J_{{}_{\max,0}},$$
(17b)$$V_{{}_{\mathrm{cmax}}}=\beta_{{}_{B}}\times V_{{}_{\mathrm{cmax},0}}$$
where *J*_max,0_ and *V*_cmax,0_ represent the unstressed values of *J*_max_ and *V*_cmax_, respectively, at the beginning of the experiment.

### 4.6. Estimation of *Γ**, R_*d*_, and C_*c*_

Katul et al. [76] and Liu et al. [77] showed that Γ* can be estimated as a function of *T*_air_:(18)$$\Gamma *=36.9+1.18\times (T_{{}_{\mathrm{air}}}-25)+0.036\times {\left(T_{{}_{\mathrm{air}}}-25\right)}^{2},$$

Here, we used air temperature as measured in the LI-6800 leaf chamber. *R*_d_ is described as [78]:(19)$$R_{{}_{d}}=0.015\times V_{{}_{\mathrm{cmax},0}},$$

Mesophyll conductance to CO_2_ was assumed to be infinite and thus *C*_c_ was considered to be equal to intercellular CO_2_ partial pressure *C*_i_, [53]. *C*_i_ is estimated as [79]:(20)$$C_{i}=C_{a}-\frac{A_{\mathrm{net}}}{G_{c}},$$
where *C*_a_ is the ambient air CO_2_ partial pressure (µmol mol^−1^), and G_c_ is the stomatal conductance for CO_2_ (mol m^−2^ s^−1^). *A*_net_ is the minimum of A_C_ and A_J_ (Equation (14)). Because *C*_i_, *A*_net_, and G_c_ are coupled to each other, the estimation of *A*_net_ and G_c_ has to be resolved iteratively over *C*_i_ given an initial value, which is *C*_i_ = 0.7 × *C*_a_ for C3 winter wheat [80]. The iterative loop stops when the difference in *C*_i_ between two successive iterations is less than 0.1 µmol mol^−1^. The biochemical model of photosynthesis proposed by Farquhar, von Caemmerer and Berry [64] was used to estimate *A*_net_ as the minimum of the Rubisco-limited CO_2_ assimilation rate and the electron-transport-limited CO_2_ assimilation rate.

G_c_ is estimated using a modified Ball–Woodrow–Berry model BWB [81]:(21a)$$G_{s}=G_{0}+\frac{a\times \beta_{{}_{S}}\times A_{\mathrm{net}}}{C_{s}\times (1+\mathrm{VPD}/D_{0})},$$
(21b)$$G_{c}=0.64\times G_{s}$$
where G_0_ is the residual conductance (mol m^−2^ s^−1^), assumed to be 0.01 [81]; *C_s_* is the CO_2_ concentration at the leaf surface (µmol mol^−1^), assumed to be the product of *a*/(*a −* 1) and *C*_i_; VPD is the vapor pressure deficit (kPa); D_0_ (kPa) is an empirical parameter related to stomatal sensitivity to VPD, assumed to be 1.5 [81]; *a* is a parameter related to *C*_i_, assumed to be 11.0 [81]; 0.64 is a factor used to convert the molecular diffusivity of water vapor to CO_2_ [79]; *β*_S_ is the normalized soil-moisture-dependent stress function which accounts for the reduction in G_S_ under water stress. *β*_S_ and *β*_B_ (Equation (17)) can be defined as:(22)$$\beta_{i}=\left\{ \begin{array}{l} \;0\;\;\;\;\;\theta_{\mathrm{SWC}}<\theta_{\mathrm{WP}}\\ {\left[\frac{\theta_{\mathrm{SWC}}-\theta_{\mathrm{WP}}}{\theta_{\mathrm{FC}}-\theta_{\mathrm{WP}}}\right]}^{q_{j}}\;\theta_{\mathrm{WP}}\leq \theta_{\mathrm{SWC}}\leq \theta_{\mathrm{FC}}\;,\\ \;1\;\;\;\;\;\theta_{\mathrm{SWC}}>\theta_{\mathrm{FC}} \end{array}\right.$$
where *β*_i_ ranges between 1 (for plants not suffering from drought) and 0 (transpiration is zero); the subscripts i = B and S are for biochemical and stomatal limitations, respectively [75]; *θ*_FC_ and *θ*_WP_ represent *θ*_SWC_ at the field capacity (24%) and wilting point (9%); the fitting parameter q_j_ is a measure of the nonlinearity of the effects of water stress on the biochemical and stomatal limitations, and j takes the values B and S for i = B and S, respectively [75,82]. q_B_ (Equation (17)) and q_S_ in (Equation (21)) were determined by minimizing a cost function, *C* = (*M − S*)^2^, where *M* represents the measurement and *S* is the corresponding simulated value (*V*_cmax_ for q_B_, and Gs for q_S_).

## 5. Conclusions

Our results show that the response of fluorescence emissions to drought is smaller than those of either stomatal conductance or net photosynthetic carbon assimilation. At the canopy scale and beyond, however, structural dynamics dominate the spatial variation of canopy SIF in response to water stress, explaining the strong drought response of SIF retrieved from space. As drought becomes more severe, the shifts in energy allocation towards decreasing photochemistry and increasing fluorescence emission tend to push plants into the PQ phase, enhancing the nonlinearity in the overall relationship between photochemistry and fluorescence. We confirm that SIF alone has a limited ability to predict drought-induced declines in photosynthetic parameters. Alternatively, the rMLR model, using SIF as one important input variable, demonstrates a satisfactory performance in reproducing declines in stomatal conductance and net photosynthetic carbon assimilation. The rMLR model has good potential for applications at regional and global scales, and thus provides the basis for using SIF mechanistically to estimate GPP under the scenario of increasing intensity and the extent of droughts in the twenty-first century.

## Figures and Tables

**Figure 1 plants-12-03365-f001:**
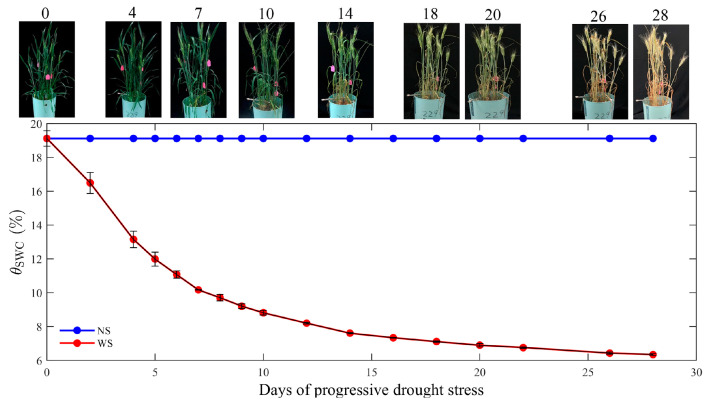
Representative images of the water-stressed winter wheat and variation in soil water content (*θ*_SWC_, %) for the non-water stress (NS, blue) and water stress (WS, red) treatments under 20 days of progressive drought stress. The numbers above the images represent days after withholding water. Datapoints and error bars represent the mean ± standard deviation (SD) of four replicates.

**Figure 2 plants-12-03365-f002:**
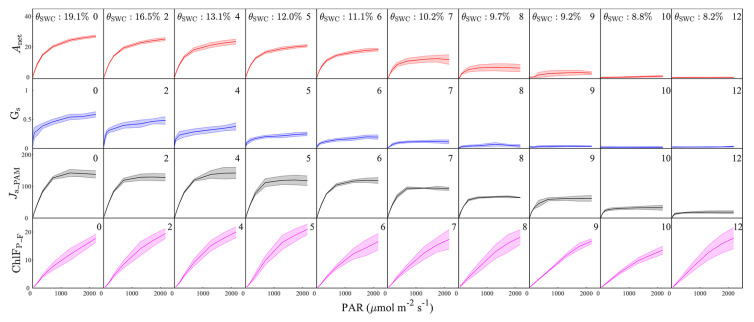
The impact of progressive drought stress on the responses of net photosynthetic carbon assimilation (*A*_net_, μmol m^−2^ s^−1^), stomatal conductance (G_S_, mol m^−2^ s^−1^), actual rate of electron transport (*J*_a_PAM_, μmol m^−2^ s^−1^), and full-band chlorophyll fluorescence emission at the photosystem level (ChlF_P_F_, μmol m^−2^ s^−1^) to changing light intensity. The number in the upper right corner of each plot is the number of days of progressive drought stress. The soil water content (*θ*_SWC_, %) is also indicated. The solid lines represent the mean, and the shaded areas are ±1 standard deviation, of four replicates. *A*_net_ and G_S_ are measured by the gas-exchange system.

**Figure 3 plants-12-03365-f003:**
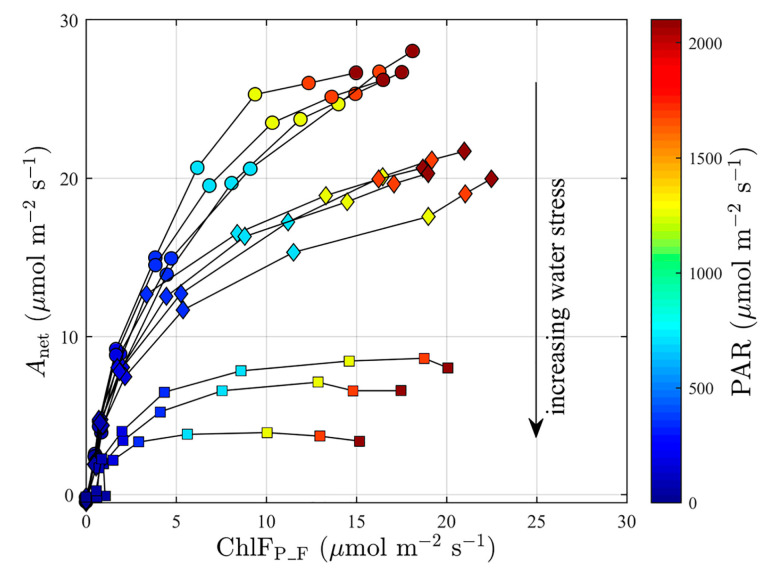
Relationships of net photosynthetic carbon assimilation (*A*_net_, μmol m^−2^ s^−1^) with full-band chlorophyll fluorescence emission at the photosystem level (ChlF_P_F_, μmol m^−2^ s^−1^). Individual light–response curves are indicated by the black lines connecting measurements obtained at increasing light levels (0–2100 μmol m^−2^ s^−1^). Circles indicate no water stress (*θ*_SWC_ = 19.1%), diamonds represent moderate water stress (*θ*_SWC_ = 12.0%), and squares indicate plants under high water stress (*θ*_SWC_ = 9.7%). The number of light–response curves under high water stress was smaller than those under no water stress and moderate water stress due to human error, leading to unrealistic values in the measurements. *A*_net_ is obtained from the gas-exchange system.

**Figure 4 plants-12-03365-f004:**
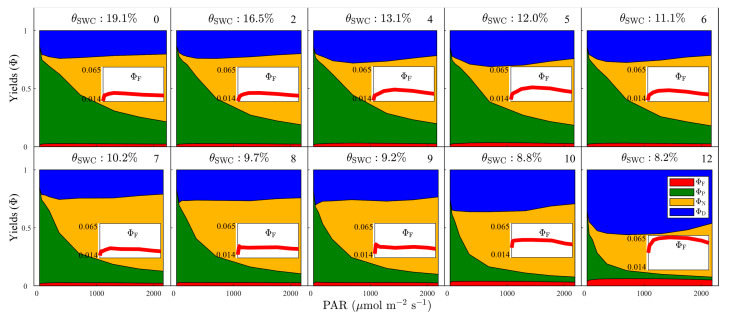
Light–response curves of the quantum yields (Ф) for the four different pathways during progressive drought stress: fluorescence (Ф_F_, red), photosynthesis (Ф_P_, green), regulated heat dissipation (Ф_N_, yellow), and basal heat dissipation (Ф_D_, blue). The number in the upper right corner of each plot indicates the day of progressive drought stress. The soil water content (*θ*_SWC_, %) of progressive drought stress treatment is also given. The inset panel illustrates the pattern of Ф_F_ under changing light intensity. The quantum yields (Ф) for the four different pathways are obtained from PAM measurements.

**Figure 5 plants-12-03365-f005:**
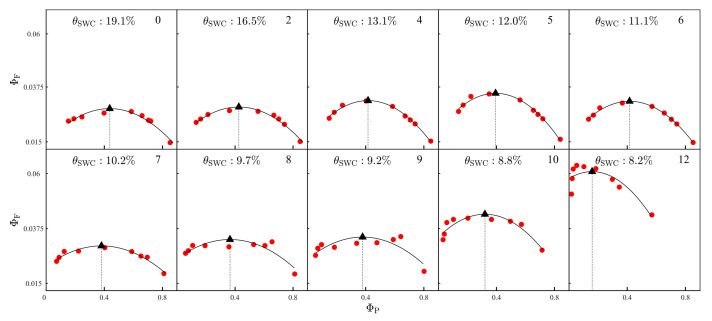
Relationships between the quantum yields of fluorescence (Ф_F_) and photochemical quenching in PSII (red circle) during the progressive onset of drought stress. Polynomial models were used to fit all relationships (black solid lines). Breakpoints (black triangles) are the value of Ф_P_ at which the slope of the Ф_P_-Ф_F_ relationship changes sign.

**Figure 6 plants-12-03365-f006:**
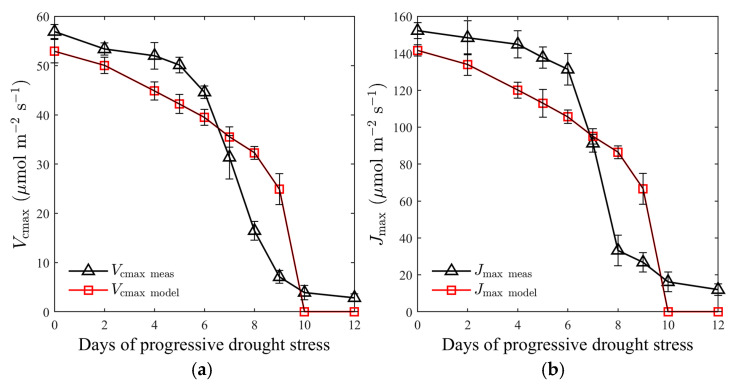
(**a**) Comparisons of measured versus simulated *V*_cmax_ (μmol m^−2^ s^−1^, *V*_cmax meas_ vs. *V*_cmax model_) between 1 and 12 days after imposing drought stress; (**b**) comparisons of measured versus simulated *J*_max_ (μmol m^−2^ s^−1^, *J*_max meas_ vs. *J*_max model_) between 1 and 12 days after imposing drought stress. Datapoints and error bars represent the mean and standard deviation of four replicates.

**Figure 7 plants-12-03365-f007:**
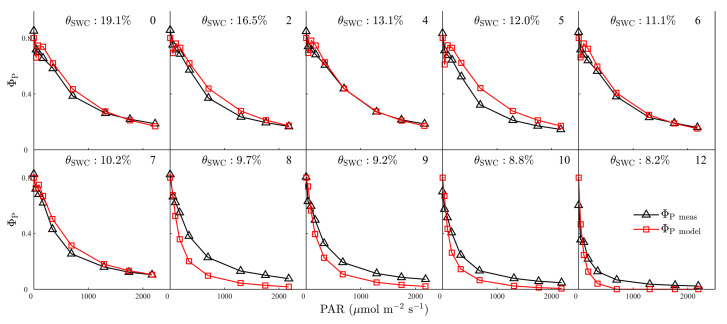
The measured versus simulated quantum yield of photochemical quenching in PSII (Ф_P_) during light–response curves between 1 and 12 days after imposing drought stress. The soil water content (*θ*_SWC_, %) of progressive drought stress treatment is indicated. The number in the upper right corner of each plot indicates the day of progressive drought stress.

**Figure 8 plants-12-03365-f008:**
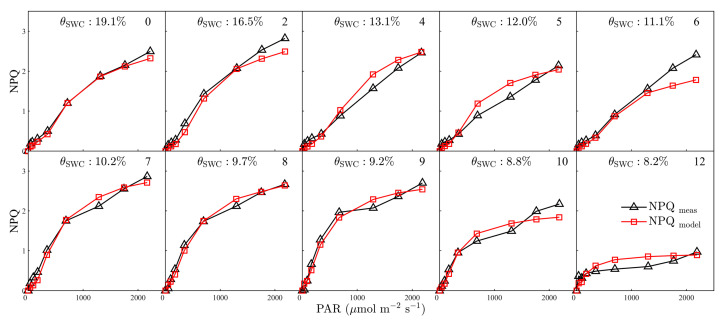
Measured versus simulated non-photochemical quenching (NPQ) during light–response curves between 1 and 12 days after imposing drought stress. The soil water content (*θ*_SWC_, %) of progressive drought stress treatment is shown. The day of progressive drought stress is indicated by the number in the upper right corner of each plot.

**Figure 9 plants-12-03365-f009:**
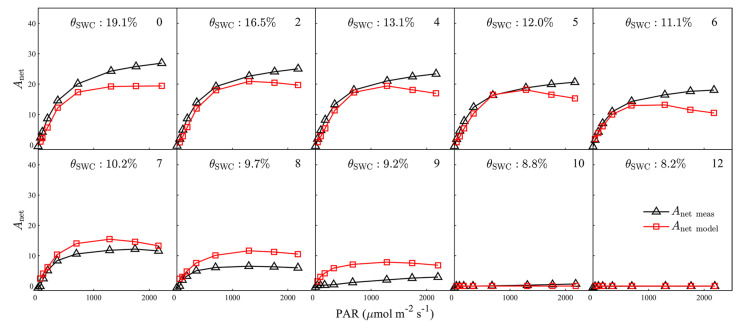
Measured versus simulated net photosynthetic carbon assimilation (*A*_net_, μmol m^−2^ s^−1^) during light–response curves between 1 and 12 days after imposing drought stress. The soil water content (*θ*_SWC_, %) of progressive drought stress treatment is indicated. The number in the upper right corner of each plot indicates the day of progressive drought stress.

**Figure 10 plants-12-03365-f010:**
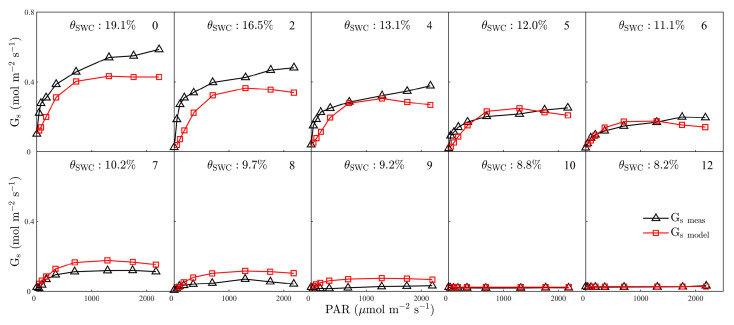
Measured versus simulated stomatal conductance (G_S_, mol m^−2^ s^−1^) during light–response curves between 1 and 12 days after imposing drought stress. The soil water content (*θ*_SWC_, %) of progressive drought stress treatment is shown. The day of progressive drought stress is indicated by the number in the upper right corner of each plot.

**Figure 11 plants-12-03365-f011:**
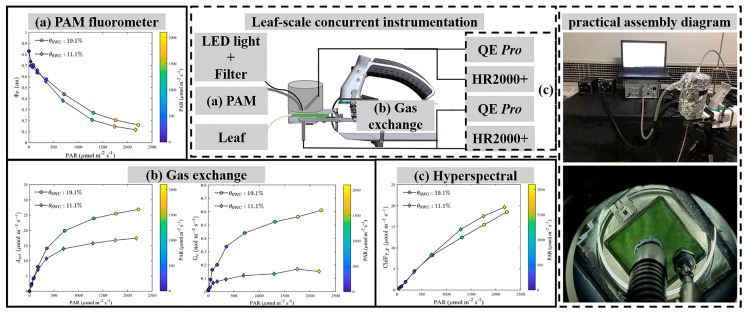
Schematic of the leaf-scale concurrent light–response curve measurement system. The following measurements were taken: (**a**) pulse amplitude modulation (PAM) fluorometer (the quantum yields of photochemical quenching in photosystem II (Ф_P_) and photochemically active radiation (PAR) are shown). (**b**) Gas exchange (net CO_2_ assimilation rate in light–response curves). (**c**) Chlorophyll fluorescence flux density emitted from photosystem II (ChlF_P_F_, μmol m^−2^ s^−1^), retrieved from the full ChlF emission spectrum. The external light source was attenuated by a customized 625 nm short-pass filter (not shown). A practical photograph of the leaf-scale concurrent measurement system which includes an LI-6800 gas-exchange chamber, a Dual-PAM-100 fluorometer, two HR2000+ spectrometers and two QE *Pro* spectrometers.

**Figure 12 plants-12-03365-f012:**
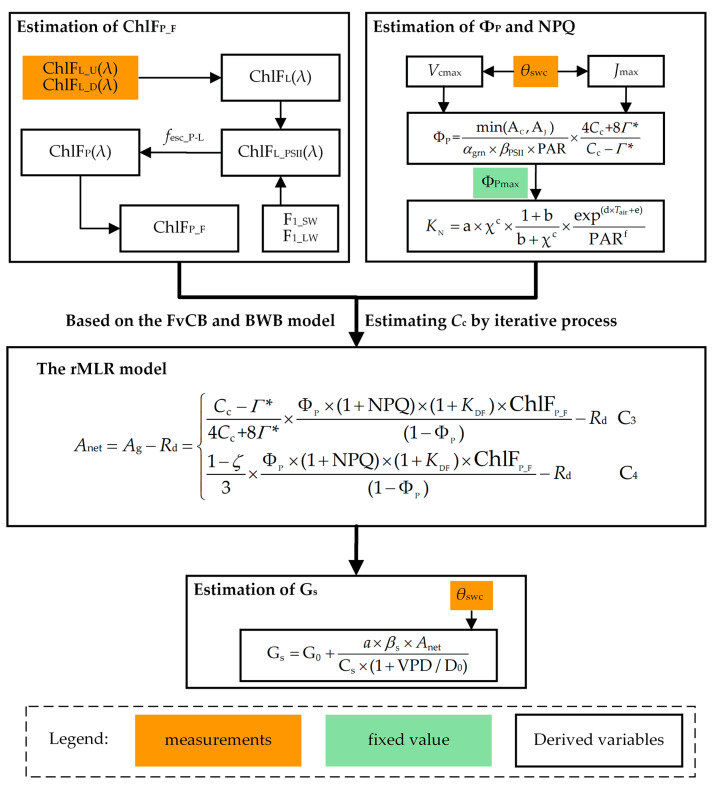
Flowchart for quantifying the net CO_2_ assimilation rate *A*_net_ (μmol m^−2^ s^−1^) from the passive ChlF spectrum. The definitions of the variables are listed in Appendix B.

## Data Availability

The data are available on request from the authors. The data supporting the findings of this study are available from the corresponding author, Xiaoliang Lu, upon request.

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
