# Peer review of "Relationship between Photosynthetic CO2 Assimilation and Chlorophyll Fluorescence for Winter Wheat under Water Stress"

_plants, 2023, doi:10.3390/plants12193365_

Round 1

Reviewer 1 Report

In my opinion the Paper is worth to be published in the present form.

Author Response

请参阅附件。

Reviewer 2 Report

This manuscripts reports on relationships between photosynthetic CO2 assimilation and chlorophyll fluorescence for winter wheat under water stress. Authors obtained valueable and original findings on relationships between photosynthetic activity, solar-induced chlorophyll fluorescence and the gross primary production during periods of drought stress. When the drought stress was more severe, authors recorded shifts in the energy allocation towards decreasing photochemistry and increasing chlorophyll fluorescence.

Authors could include into their disccusion also some aspects of stomata  behavior and jasmonate signalling and volation emissions as recorded by other authors under plant stress.

Plant Signaling & Behavior, 16:7, 1917169, DOI: 10.1080/15592324.2021.1917169

Int. J. Mol. Sci. 2020, 21, 1018; doi:10.3390/ijms21031018

Plants 2021, 10, 485. https:// doi.org/10.3390/plants10030485

This manuscripts reports on relationships between photosynthetic CO2 assimilation and chlorophyll fluorescence for winter wheat under water stress. Authors obtained valueable and original findings on relationships between photosynthetic activity, solar-induced chlorophyll fluorescence and the gross primary production during periods of drought stress. When the drought stress was more severe, authors recorded shifts in the energy allocation towards decreasing photochemistry and increasing chlorophyll fluorescence.

Authors could include into their disccusion also some aspects of stomata  behavior and jasmonate signalling and volation emissions as recorded by other authors under plant stress.

Plant Signaling & Behavior, 16:7, 1917169, DOI: 10.1080/15592324.2021.1917169

Int. J. Mol. Sci. 2020, 21, 1018; doi:10.3390/ijms21031018

Plants 2021, 10, 485. https:// doi.org/10.3390/plants10030485

Author Response

请参阅附件。

Reviewer 3 Report

The study provides an insight on the Relationship between photosynthetic CO2 assimilation and chlorophyll fluorescence for winter wheat under water stress. The study findings provide experimental and theoretical foundations necessary for employing SIF mechanistically to estimate plant photosynthetic activity during periods of drought stress. However, there are some limitations which must be addressed.

In title write Co2 correctly.

The abstract is very general, addition of some specific results such as quantitative results are recommended.

Clearly and briefly mention the techniques used in this study in the abstract.

Introduction should reflect the title such as provide relationship between photosynthesis CO2 assimilation and chlorophyll fluorescence.

Effects of winter of temperature on photosynthesis CO2 assimilation and chlorophyll fluorescence.

Impacts of water stress on photosynthesis CO2 assimilation and chlorophyll fluorescence and also response of wheat.

Based on the above points provide justification statement and hypothesis of the study in the introduction.

Line 55-57 should be cited with some recent studies  doi: 10.1039/d2gc02467e, https://doi.org/10.3390/ijms22179175

While the concurrent measurements of active and passive fluorescence, gas-exchange rates, and other environmental parameters are commendable, it is crucial to consider potential limitations or sources of error in the measurement techniques employed. Any uncertainties or potential biases in the measurement methods should be acknowledged and discussed in the discussion section.

The study does not provide explicit information on the sample size or the statistical analysis performed. Reporting these details would enhance the transparency and robustness of the results.

It would be valuable to assess the statistical significance of the observed relationships and quantify the uncertainty associated with the estimated parameters.

The assumptions and limitations of the model should be clearly stated and discussed.

Sensitivity analyses or validation against independent datasets could help assess the model's accuracy and reliability.

The study briefly mentions mixed results in previous studies on the impact of drought on the SIF-GPP relationship. A more comprehensive discussion and comparison with existing literature would help contextualize the findings and contribute to the current understanding of the topic.

Addressing the challenges of implementing the proposed SIF-based model in real-world scenarios, such as field-scale measurements or remote sensing applications, would enhance the study's relevance. 

Avoid long sentences such as line 70-73

Author Response

请参阅附件。

Round 2

Reviewer 3 Report

All corrections are made and the MS can be accepted